# Nominal elastic modulus assessment in 3D-printed components under varying printing parameters using Bayesian methods and random forest surrogate modeling

Jin Zhang[1], Lili Lu[2]*, Ping Feng[3], Ting Zhu[4]

**1** College of Art Design, The College of Post and Telecommunication of WIT, Wuhan, Hubei, China, **2** Wuhan Huaxia University of Technology, Wuhan, China, **3** Wuhan Police Vocational College, Wuhan, China, **4** College of Art Design, The College of Post and Telecommunication of WIT, Wuhan, Hubei, China

* luliliwhi@163.com

## Abstract

The expanding range of materials available for 3D printing is driving its widespread adoption in advanced fields. As 3D printing becomes increasingly prevalent in the manufacturing of industrial components, its advantages in accommodating complex geometries and reducing material waste are attracting significant attention. Acquiring and applying precise elastic properties of materials during structural design is crucial for ensuring part safety and consistency. However, non-destructive mechanical property assessment methods remain limited. In this paper, we propose an efficient surrogate model, built using a Bayesian model updating approach combined with a random forest algorithm, to achieve high-precision calibration of material elastic constants. In the experiment, samples were 3D printed using fused deposition modeling, and modal information was obtained using operational modal analysis with one end fixed to simulate cantilever beam boundary conditions. Parameter updating was then performed within a Bayesian Markov Chain Monte Carlo framework. The deviation between the updated calculated frequencies and the measured frequencies was significantly reduced, and the Modal Assurance Criterion value between the updated calculated mode shapes and the measured mode shapes was higher than 0.99, demonstrating the accuracy of the updated parameters. Compared to traditional destructive testing methods, the proposed method directly calibrates the structural elastic modulus at the component level without affecting the normal use of the component, providing a more practical approach for the analysis and research of material properties in 3D printing additive manufacturing. The related technology can be extended to other structural forms of 3D-printed products.

**Data availability statement:** All relevant data are within the manuscript and its Supporting Information files.

**Funding:** The author(s) received no specific funding for this work.

**Competing interests:** The authors have declared that no competing interests exist.

## 1. Introduction

3D printing has significantly advanced the industrial lightweighting process through its capabilities in free design, topological optimization, and the formation of complex structures [1]. Due to its highly adaptable manufacturing capabilities and ease of use, 3D printing has found widespread application in the rapid prototyping of complex geometries in metals, polymers, and fiber-reinforced composites [2]. Plastic filament is melted by heating and then layer by layer deposited through a nozzle, gradually accumulating to form the finalized shape. This technology features advantages such as fast processing speeds, low costs, and simple equipment, making it widely used for the manufacturing of functional and structural components [3].

Fused Deposition Modeling (FDM) is one of the most representative rapid prototyping methods within 3D printing [4]. The FDM process can achieve low-cost preparation of high-performance composite materials while reducing subsequent processing steps and offering good recyclability and reprocess ability. However, the forming accuracy of FDM components is largely determined by the precision of the FDM machine itself, and errors can occur at each stage of the manufacturing process, affecting the final precision [5]. Recently, numerous studies have evaluated the impact of FDM process parameters such as layer thickness [5], nozzle temperature [6], infill density [7], and printing speed [8] on the mechanical properties of manufactured parts. These studies have found that these factors significantly influence the strength of the produced components.

Due to the influence of interlayer adhesion, void defects, and various coupled effects of process parameters during the manufacturing of FDM components, obtaining accurate elastic parameters is essential for subsequent structural safety assessments and optimization designs. Currently, the commonly used methods for characterizing mechanical properties predominantly rely on destructive testing, such as uniaxial tensile tests [9], compression tests [10], shear tests [11], and fatigue tests [12]. However, these methods not only damage the test specimens but also require the preparation of samples that conform to standard dimensions. For components that are limited in volume, scarce in samples, or already formed functional parts, it is often difficult to conduct repeated testing and obtain information regarding the internal anisotropy of the components.

To overcome these limitations, recent years have seen a diverse range of research focused on the calibration of material parameters for 3D printed materials and components. Lesueur et al. [13] utilized numerical modeling to predict the yield strength of porous materials and examined the impact of internal geometries using an elastoplastic model. Luo et al. [14] conducted static and dynamic experiments on 3D printed components made of PLA and PLA-Cu, simulating their stress–strain curves. Somireddy et al. [15] proposed a constitutive model for 3D printing based on tensile test data to perform numerical simulations of elastic material properties. Khosravani [16] conducted a series of static tensile tests to analyze the effects of raster orientation and printing speed on the mechanical properties of materials. Lißner et al. [17] established a framework combining experimental and numerical methods, using experimental data from solid 3D printed compression specimens to calibrate material

models, and validated the accuracy of the calibrated models using samples of different geometries. Ji et al. [18] further investigated the static and dynamic mechanical properties and constitutive models of PLA and PLA-Cu materials under different loading rates. Gallup et al. [19] and Mencarelli et al. [20] quantified the sensitivity of mechanical behavior to key FDM printing parameters for thermoplastic polyurethane and PLA, respectively. These advancements have greatly facilitated progress in the parameter calibration of 3D printed components. However, some shortcomings remain. For instance, most studies primarily focus on identifying various mechanical parameters at the material level, lacking parameter updating methods based on finite element model updating at the component level to accurately quantify structural mechanical parameters. Moreover, many works inadequately consider the inherent randomness of materials and the uncertainty of testing, resulting in relatively singular identification outcomes [18–20].

Similar challenges are also widespread in the fields of civil engineering and aerospace. In these domains, researchers often employ finite element model updating methods for parameter inference and performance prediction, as some critical parameters are difficult to measure directly [21–23]. However, traditional model updating methods typically rely on deterministic assumptions, making it challenging to comprehensively quantify uncertainties in both the model and observations. In recent years, Bayesian model updating methods have gained widespread attention for their ability to systematically integrate prior knowledge with observational data while naturally handling uncertainties, providing a more robust and flexible approach for parameter identification and model updating [21–23].

Manufacturing parts with stable mechanical properties has always been a challenging task in FDM technology; therefore, obtaining accurate material mechanical parameters is particularly important for predicting the mechanical performance of 3D printed structures. Based on this, the present study intends to introduce a Bayesian model updating method based on Markov Chain Monte Carlo (MCMC) sampling, combined with finite element modeling, to achieve precise identification and calibration of elastic moduli for 3D printed materials, thereby enhancing the accuracy and reliability of mechanical performance predictions for 3D printed components. Considering that the Bayesian MCMC method requires multiple runs of complex finite element models, resulting in high computational costs, this study incorporates advanced random forest algorithms [24–25] from machine learning to replace time-consuming finite element reanalysis, achieving efficient mapping between structural parameters (inputs) and structural responses (outputs) to enhance the reliability and application value of 3D printed material performance predictions.

This paper is organized as follows: Section 2 presents the theoretical background, beginning with the principles of the Bayesian method in Section 2.1, followed by an exploration of the Random Forest-Based Surrogate Model (RFSM) in Section 2.2, and concluding with the methodology workflow in Section 2.3. In Section 3, the preparation of nine components using various materials and properties through 3D printing is detailed, alongside dynamic testing conducted to obtain measured modal data. A RFSM is then trained on these data, and Bayesian methods are applied for model updating. Finally, Section 4 offers the conclusions drawn from this study.

## 2. Theory

### 2.1 Bayesian method

The Bayesian method [26] is an approach grounded in probability that deduces the posterior PDF of structural parameters by integrating actual observational data with prior knowledge. This method relies on utilizing a likelihood function constructed from observational data, alongside prior information about the parameters, to compute the posterior PDF of those parameters. Following the standard likelihood formulation for modal-data model updating [27], the likelihood can be written as:

$$p(\mathbf{D}|\theta) \propto \exp\left(-\frac{1}{2}\sum_{k=1}^{n}\left[(\omega_k^m - \omega_k^c(\theta))^T \sum_{\omega}^{-1}(\omega_k^m - \omega_k^c(\theta)) + \sum_i (\phi_{i,k}^m - \phi_{i,k}^c(\theta))^T \sum_{\phi,i}^{-1}(\phi_{i,k}^m - \phi_{i,k}^c(\theta))\right]\right)$$

(1)

where $\omega_k^m$ and $\omega_k^c(\theta)$ denote the measured and computed frequency vectors, respectively. The variables $\phi_{i,k}^m$ and $\phi_{i,k}^c(\theta)$ represent the measured and computed mode shapes of the $i$-th order. The matrices $\sum_\omega$ and $\sum_{\phi,i}$ are the covariance matrices of the measured frequencies and mode shapes, while $k$ signifies the $k$-th measured mode.

Under the Gaussian-error assumption, this expression can be simplified as:

$$L(\boldsymbol{\theta}) = \exp\left(-\frac{1}{2}\boldsymbol{r}^T\Sigma^{-1}\boldsymbol{r}\right)$$

(2)

where $\boldsymbol{r}$ denotes the residual vector between measured and computed data.

When considering both frequency and mode shape measurements, the residual can be explicitly denoted as:

$$\boldsymbol{r} = \begin{bmatrix} \omega_k^m - \omega_k^c(\boldsymbol{\theta}) \\ \phi_{i,k}^m - \phi_{i,k}^c(\boldsymbol{\theta}) \end{bmatrix}$$

(3)

Given the complexity of directly computing the posterior PDF, this study utilizes Bayesian MCMC techniques [28] for sampling. These techniques streamline the process of obtaining the posterior PDF of the damage parameters. We follow the Metropolis–Hastings (MH) procedure, whose fundamental steps are summarized in [29], as follows:

1. Initialize the parameter vector $\boldsymbol{\theta}^{(t)}$, $(t = 0)$.

2. Generate a candidate sample $\boldsymbol{\theta}^*$ using the proposal distribution $q(\boldsymbol{\theta}^*|\boldsymbol{\theta}^{(t)})$.

3. Calculate the acceptance ratio:

$$\alpha = \min\left(1, \frac{p(\boldsymbol{D}|\boldsymbol{\theta}^*)p(\boldsymbol{\theta}^*)}{p(\boldsymbol{D}|\boldsymbol{\theta}^{(t)})p(\boldsymbol{\theta}^{(t)})}\frac{q(\boldsymbol{\theta}^{(t)}|\boldsymbol{\theta}^*)}{q(\boldsymbol{\theta}^*|\boldsymbol{\theta}^{(t)})}\right)$$

(4)

4. Accept or reject the candidate:

$$\boldsymbol{\theta}^{(t+1)} = \begin{cases} \boldsymbol{\theta}^*, & \text{with probability } \alpha \\ \boldsymbol{\theta}^{(t)}, & \text{otherwise} \end{cases}$$

(5)

5. Repeat steps 2–4 until convergence is achieved.

In this study, the Bayesian MCMC framework is employed to infer the nominal elastic modulus of 3D printed components by aligning computational predictions with experimental modal data. A significant practical challenge arises from the inherent nature of the MCMC sampling process. As previously mentioned, each step in the Markov chain requires the evaluation of the likelihood function (Equation 2), which in turn necessitates a complete finite element (FE) analysis to compute the modal characteristics ($\omega_k^c(\boldsymbol{\theta})$ and $\phi_{i,k}^c(\boldsymbol{\theta})$ given a candidate parameter set $\boldsymbol{\theta}^*$ [30]. Considering that convergence to a posterior distribution often necessitates thousands or even millions of such iterations, directly coupling MCMC with finite element models incurs prohibitively high computational costs. To circumvent this issue and make the inference process computationally feasible, this study introduces a surrogate model. This surrogate model serves as a low-cost yet highly accurate approximation of the FE model, enabling rapid computation of the required modal responses during the MCMC iterations [31], thereby significantly accelerating the parameter updating process.

## 2.2 Random forest-based surrogate model

To address the high computational costs associated with directly coupling the MCMC method with FE analysis discussed in Section 2.1, this study employs a surrogate model based on the Random Forest (RF) algorithm. The RF algorithm is

a powerful ensemble learning method that approximates complex nonlinear mappings between inputs and outputs by combining multiple regression trees [24,25]. By learning from a pre-computed set of high-fidelity FE simulation results, the trained RF model can provide instantaneous predictions of structural modal characteristics, serving as an efficient computational surrogate for the complete FE model.

In the context of this study, the RFSM is trained to approximate the true function $f$ that maps material parameters to structural responses:

$$y = f(\mathbf{x}) + \varepsilon \tag{6}$$

where $\varepsilon$ represents inherent noise or model error. Specifically, the input vector $\mathbf{x}$ corresponds to the uncertain parameters $\boldsymbol{\theta}$ in the Bayesian framework, which, in this study, is the nominal elastic modulus $\mathbf{E}$. The output $y$ represents the corresponding modal data—specifically, the natural frequencies and mode shape components computed from the FE model.

The predicted value from the RF, $\hat{f}_{\text{RF}}(\mathbf{x})$, is given by the average predictions of a large number of independent regression trees:

$$\hat{f}_{\text{RF}}(\mathbf{x}) = \frac{1}{B} \sum_{b=1}^{B} T_b(\mathbf{x}) \tag{7}$$

where $B$ is the total number of trees, and $T_b(\mathbf{x})$ is the prediction from the $b$-th tree. To ensure diversity among the trees and mitigate overfitting, each tree $T_b$ is trained on a bootstrap sample $\mathcal{D}_b$, randomly drawn with replacement from the original training dataset $\mathcal{D} = \{(\mathbf{x}_i, y_i)\}_{i=1}^{N}$:

$$\mathcal{D}_b \sim \text{Bootstrap}(\mathcal{D}) \tag{8}$$

At each split node of the tree, RF enhances randomness by considering only a random subset of input features $\mathcal{F}_b \subseteq \{1, 2, \ldots, d\}$. The optimal split is determined by minimizing an impurity measure, typically the mean squared error (MSE):

$$\text{MSE}(S) = \frac{1}{|S|} \sum_{i \in S} (y_i - \bar{y}_S)^2 \tag{9}$$

where $S$ is the set of samples in a node and $\bar{y}_S$ is the average output in $S$. The best split $(j^*, t^*)$ is defined as the one that maximally reduces the MSE:

$$(j^*, t^*) = \arg\min_{j \in \mathcal{F}_b, t} \left( \frac{|S_L(j, t)|}{|S|} \text{MSE}(S_L) + \frac{|S_R(j, t)|}{|S|} \text{MSE}(S_R) \right) \tag{10}$$

where $S_L$ and $S_R$ are the left and right child nodes after splitting on feature $j$ at threshold $t$. Each tree $T_b$ partitions the input space into $M_b$ disjoint regions $\{R_{m,b}\}_{m=1}^{M_b}$ and predicts a constant $c_{m,b}$ within each region:

$$T_b(\mathbf{x}) = \sum_{m=1}^{M_b} c_{m,b} I(\mathbf{x} \in R_{m,b}) \tag{11}$$

where $I(\cdot)$ is the indicator function. The constant $c_{m,b}$ is typically the average of training outputs in region $R_{m,b}$

$$c_{m,b} = \frac{1}{|R_{m,b}|} \sum_{i \in R_{m,b}} y_i \tag{12}$$

Once the training is complete, the RFSM provides a high-fidelity approximation of the true FE model:

$$\hat{f}_{\text{RF}}(\boldsymbol{x}) \approx f(\boldsymbol{x})$$

(13)

Enabling rapid computation of the frequencies and mode shapes required for likelihood function evaluations in the MCMC algorithm. Additionally, the predictive variance of this ensemble model can also be estimated to quantify the model's own uncertainty:

$$\hat{\sigma}^2(\boldsymbol{x}) = \frac{1}{B-1} \sum_{b=1}^{B} \left( T_b(\boldsymbol{x}) - \hat{f}_{\text{RF}}(\boldsymbol{x}) \right)^2$$

(14)

This makes the entire Bayesian parameter updating process for identifying the elastic modulus of 3D printed components computationally feasible.

## 2.3 Methodology workflow

This study integrates a Bayesian inference framework with a RFSM to achieve efficient identification of material parameters for 3D printed components. This hybrid strategy capitalizes on the robust capabilities of Bayesian methods in quantifying uncertainty while leveraging the surrogate model to overcome the computational bottlenecks associated with repeated calls to FE models in traditional Bayesian updating. The overall workflow (as illustrated in Fig 1) is systematic and logical, providing a coherent pathway for extracting true material properties from experimental data. The detailed technical steps can be summarized as follows:

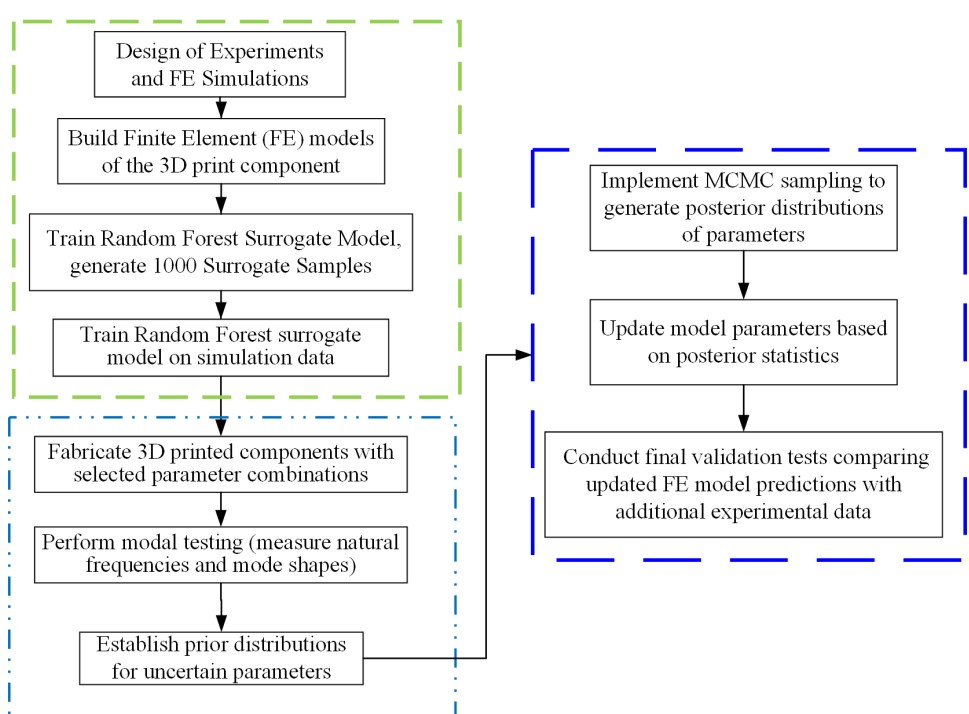

**Fig 1. Overall workflow of the proposed framework.**

First, define the parameter space and generate training data. The core parameter, namely the nominal elastic modulus $\mathbf{E}$, is established to have a reasonable range based on the physical properties of the material and prior knowledge. Subsequently, design methods such as Latin Hypercube Sampling [43] are employed to select $N$ design points $\{\mathbf{E}_i\}_{i=1}^N$ within this parameter space. For each design point, a high-fidelity FE analysis is conducted to compute the corresponding structural dynamic responses (i.e., natural frequencies and mode shapes), thus constructing a complete input-output training dataset $(\mathbf{E}_i, \mathbf{y}_i)_{i=1}^N$.

Utilizing the dataset generated above, and employing Eqs. (6) through (14), a RFSM model is trained to learn and approximate the complex nonlinear relationship $\hat{f}_{RF} : E \mapsto \mathbf{y} = (\omega^c, \phi^c)$ from input (material parameters) to output (structural dynamic responses). Once the surrogate model is trained, it can instantaneously predict the structural frequency and mode shape corresponding to any specified elastic modulus $\mathbf{E}$.

Next, we define the likelihood function $p(\mathbf{D}|\theta)$ (Eq. (1)) to quantify the consistency between the model predictions and experimental measurements. Here, $\mathbf{D} = \mathbf{y}_{obs}$ represents the observed data. In this context, the well-trained RFSM model $\hat{f}_{RF}(\theta)$ is utilized to replace the actual FE model.

Next, MCMC sampling is conducted using Eqs. (2) through (5). To identify the elastic modulus $\mathbf{E}$, a suitable prior distribution $p(\theta)$ is selected to represent the initial understanding before any experimental data is observed. Subsequently, the Metropolis-Hastings (MH) algorithm [32] is employed to perform the MCMC sampling. In each iteration, the MH algorithm proposes a candidate parameter and utilizes the RFSM model (instead of the FE model) to rapidly compute the likelihood function. The acceptance criteria are then applied to determine whether to accept the proposed parameter.

Upon convergence of the Markov chain, a substantial number of samples from the posterior probability distribution of the parameters are obtained. This results in the posterior PDF of the elastic modulus $\mathbf{E}$. Analyzing this PDF (for example, by calculating its mean or peak value) allows for the determination of the optimal estimate of the identified parameter and facilitates the quantification of its uncertainty (such as establishing confidence intervals).

In summary, this methodology effectively integrates experimental data with physical models within the Bayesian framework while employing surrogate modeling techniques to address computational efficiency challenges. Ultimately, it enables precise identification of the nominal elastic modulus $\mathbf{E}$ for 3D printed components, thereby providing a robust foundation for subsequent mechanical performance predictions, structural health monitoring, and optimizing design of such components.

## 3 Component 3D printing and parameter updating

### 3.1 Component design and printing

The working principle of the 3D printer using FDM is illustrated in Fig 2. The system is divided into four main zones based on their functions: the feeding zone, cooling zone, melting zone, and forming zone, with the feeding zone dedicated to the extrusion of filament. Together, these zones contribute to the complete printing of the workpiece.

To systematically investigate the impact of material types and key process parameters on the mechanical properties of additive manufacturing components, an orthogonal experimental design was developed. The experimental subjects consist of uniform beam specimens with dimensions of 250 mm × 15 mm × 15 mm. The study primarily examines two core factors:

**1. Material Type**: Three representative thermoplastics commonly used in FDM were selected:

**Polylactic Acid (PLA)** [33]: Widely recognized as the most commonly used 3D printing material, PLA is noted for its excellent stiffness, ease of printing, and high forming precision.

**Polyethylene Terephthalate Glycol (PETG)** [34]: This material exhibits superior overall performance compared to PLA, with enhanced toughness, impact resistance, and chemical corrosion resistance.

**Acrylonitrile Butadiene Styrene (ASA)** [35]: An engineering-grade plastic that has mechanical strength comparable to ABS, ASA stands out for its exceptional UV resistance and weatherability, making it ideal for functional applications and outdoor parts.

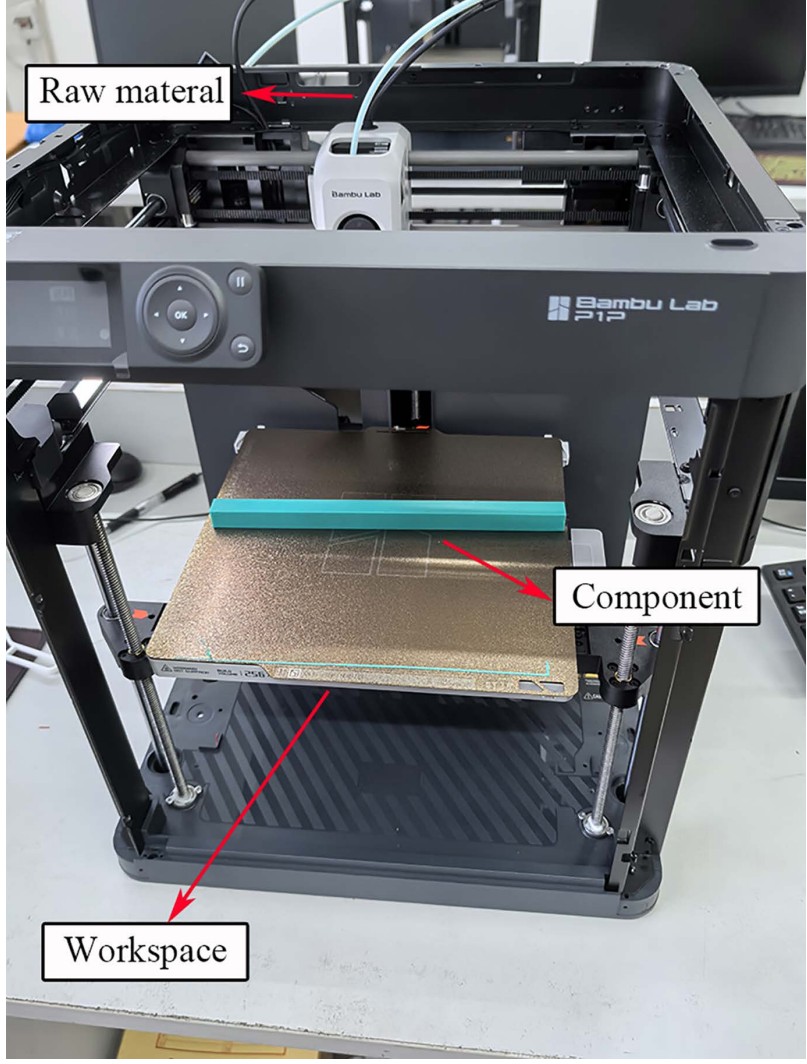

**Fig 2. Printing equipment for specimen fabrication.**

**Infill Density**: This parameter defines the sparsity of the internal filling structure of the printed component and is a key process parameter that determines the overall strength, stiffness, and weight of the component. The study established three significant levels: 5%, 15%, and 30%.

Through conducting orthogonal experiments with the aforementioned combinations of materials and infill densities, this study aims to comprehensively evaluate the performance differences of the specimens under various configurations. The complete experimental configuration matrix provides detailed information on the specific parameter combinations for each specimen, including PLA specimens (1–3), PETG specimens (4–6), and ASA specimens (7–9), as shown in Table 1.

The 3D printing process using FDM was conducted according to the parameters set in Table 1. Prior to printing, the slicing software formatted the three-dimensional Computer Aided Design (CAD) model of the physical workpiece into Standard Template Library (STL) format, outputting G-code to the control software. The extruder then returned to the origin as specified by the limit switches in accordance with the G-code instructions.

**Table 1. Printing parameter configuration.**

| No. | Material | Infill Density | Print Speed(mm/s) | | | | | | | Mass (g) |
| | | | First Layer | First Layer Fill | Outer Wall | Inner Wall | Sparse Fill | Solid Fill | Top Surface | |
|---|---|---|---|---|---|---|---|---|---|---|
| 1 | PLA | 5% | 50 | 105 | 200 | 300 | 270 | 250 | 200 | 17.8 |
| 2 | | 15% | 50 | 105 | 200 | 300 | 270 | 250 | 200 | 21.8 |
| 3 | | 30% | 50 | 105 | 200 | 300 | 270 | 250 | 200 | 30 |
| 4 | PETG | 5% | 50 | 105 | 200 | 300 | 270 | 250 | 200 | 18.3 |
| 5 | | 15% | 50 | 105 | 200 | 300 | 270 | 250 | 200 | 22.9 |
| 6 | | 30% | 50 | 105 | 200 | 300 | 270 | 250 | 200 | 31.1 |
| 7 | ASA | 5% | 50 | 105 | 200 | 300 | 270 | 250 | 200 | 15.5 |
| 8 | | 15% | 50 | 105 | 200 | 300 | 270 | 250 | 200 | 20.1 |
| 9 | | 30% | 50 | 105 | 200 | 300 | 270 | 250 | 200 | 26.3 |

Subsequently, the thermoresistor began heating the working platform and the hot end of the extruder to the designated temperature. The filament was driven to the cold end by the drive gear and idler gear, where it was then heated and melted before being extruded through the nozzle onto the heated bed. The nozzle diameter was set to 0.2 mm and the layer height to 0.2 mm. The nozzle temperature was maintained at 220 °C, with an average room temperature of 26 °C. Once melted, the filament accumulated in lines on the heated bed platform, referred to as paths. The deposition of multiple paths formed a single layer of the workpiece. After each layer was completed, the extruder moved upward by a specified Z value or layer height, repeating this process for subsequent layers until the entire workpiece was printed. The finished components are displayed in Fig 3.

Fig 4 presents detailed views of the internal infill grid structures within the printed components. As shown in Fig 4a and 4b, the specimens with 15% and 30% infill densities exhibit square grid patterns with dimensions of 3 mm and 6 mm, respectively, which are consistent with the grid configurations specified during printing. However, the component with 5% infill density in Fig 4c displays incomplete square patterns due to the constraints imposed by the low infill density setting. Fig 4d, 4e, and 4f provide scanning electron microscopy (SEM) images captured along the infill direction for the three components shown in Fig 4. These SEM images reveal that the actual layer height closely matches the preset printing layer thickness of approximately 0.2 mm across all specimens. The layer stacking appears relatively uniform throughout, with PETG material demonstrating the most consistent and orderly layering arrangement. These observations indicate that the printer's process parameters were well-controlled during fabrication.

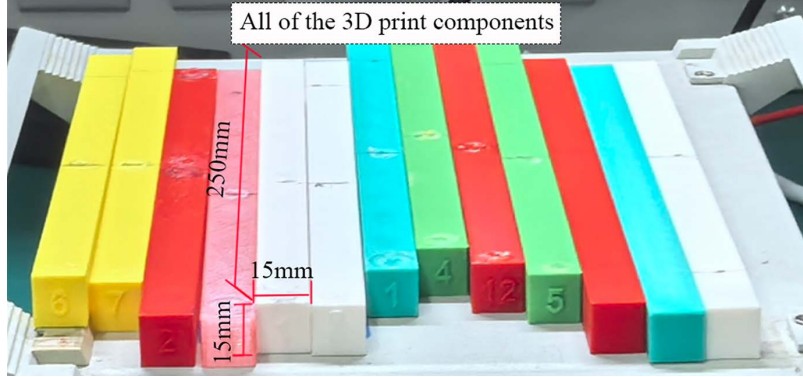

**Fig 3. Display of printed components.**

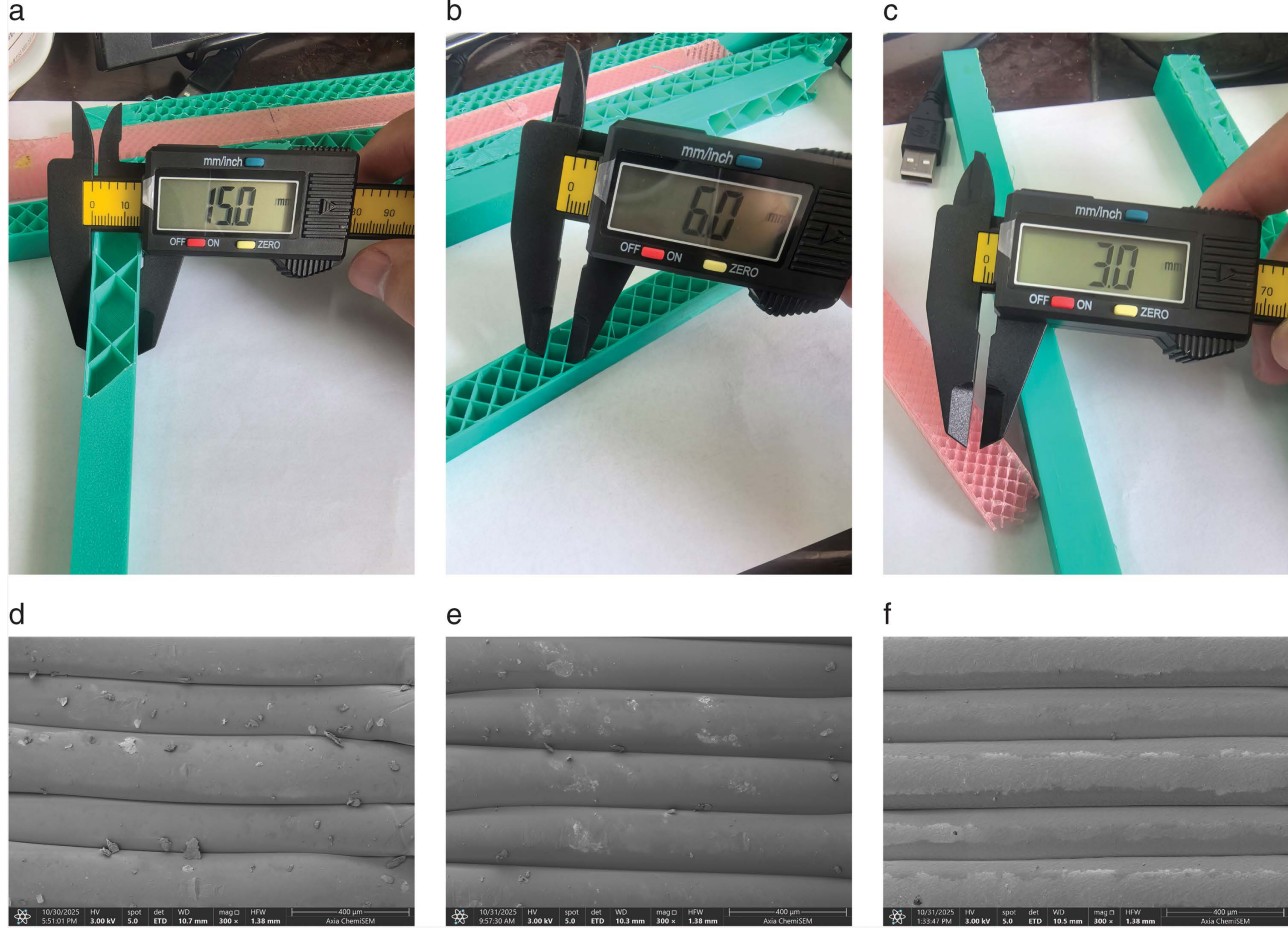

**Fig 4. Dimensional checks and internal morphology of FDM specimens at three infill densities. (a)** PLA specimen printed at 5% infill; **(b)** ASA specimen at 15% infill; caliper measurement; **(c)** PETG specimen at 30% infill; **(d)** SEM of PLA, 5% infill; **(e)** SEM of ASA, 15% infill; **(f)** SEM of PETG, 30% infill.

### 3.2 Modal testing of components

Modal testing will be conducted on the components to inverse the elastic modulus of the materials. The modal testing will employ the operational modal analysis (OMA) method, utilizing the obtained frequency and mode shape response data. The elastic modulus for each specimen will be determined using Eq. (2). One end of the specimen will be fixed with a clamp at 30 mm to simulate a cantilever beam. The length of the cantilever portion is 22 cm, and two accelerometers will be positioned at both the midspan and the cantilever end of the specimen to collect the vertical acceleration response during vibration, as illustrated in Fig 5. The sampling frequency is set to 10 kHz. The sensors will be adhered to the specimen using high-strength adhesive.

Prior to testing, the amplitude and phase of the accelerometers will be carefully calibrated to ensure the accuracy and consistency of data collection. Environmental excitation will be simulated using random tapping, with dynamic data acquisition performed through the Yutai dynamic signal acquisition and control testing system. After data collection, the Experimental Modal Analysis (EMA) using the Expanded Frequency Domain Decomposition (EFDD) method [36] will be applied to calculate the measured frequencies and mode shapes, which will be compared with the results of finite element simulations. The measured first two modal frequencies are presented in Table 2.

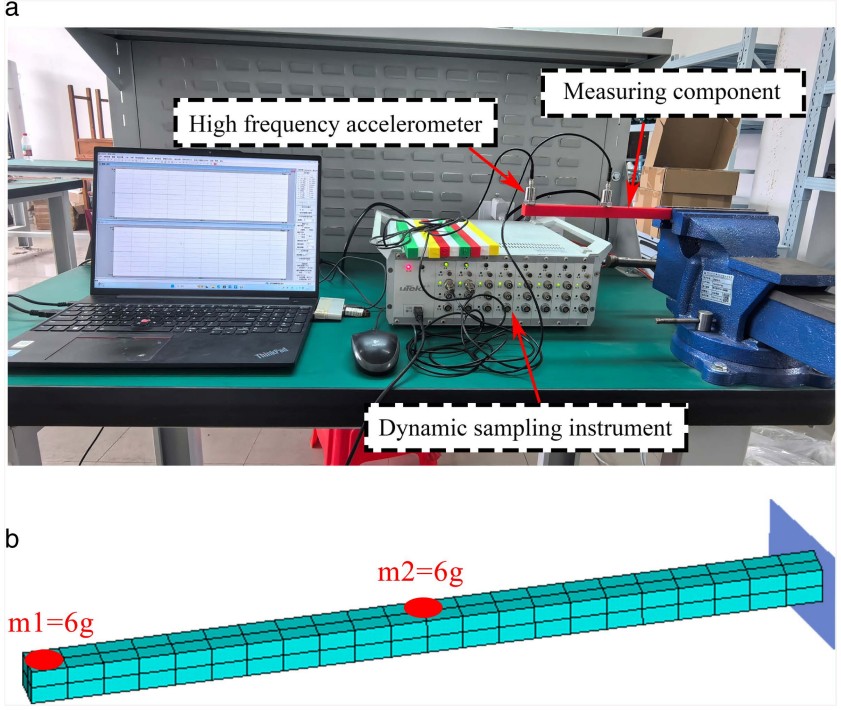

**Fig 5. Experimental setup and FE model for specimen. (a)** Accelerometer arrangement for impact testing, **(b)** The Beam FE mesh including two 6g masses at the tip (m1) and near mid-span (m2).

It is noted that, due to the uncertain initial nominal elastic modulus, all components have their elastic modulus initially estimated at 15% of the elastic modulus under full infill conditions, which is valued at 4000 MPa [37], resulting in a preliminary estimate of $E_0 = 600$MPa.. As shown in Fig. 5(b), the FE model of the specimen was established in ANSYS using the BEAM188 element. BEAM188 is a three-dimensional beam element suitable for simulating bending, shear, and torsional deformations. The beam has a square cross-section of 1.5 cm × 1.5 cm, with a total length of 25 cm. The rightmost 3 cm of the beam was clamped to represent the fixture, thereby applying a fixed boundary condition at the nodes of this region. The beam was discretized along its longitudinal axis into 22 elements, each 1 cm in length. The mass of each segment was assigned according to the actual mass values listed in Table 1, assuming a uniform distribution. In addition, the mass of the sensor and its connecting cable (6 g) was incorporated into the initial model as a lumped mass through the MASS21 element, thereby accounting for the extra inertia introduced by the measurement system.

### 3.3 Modal testing results

The results in Table 2 show that the calculated first two modal frequencies of components fabricated from three materials (PLA, PETG, ASA) decrease markedly with increasing infill density. This behavior can be explained using the dynamic relation for natural frequencies, $f \propto \sqrt{E/M}$, where $E$ is the elastic modulus and $M$ is the structural mass (see Table 1). Since the nominal elastic modulus used in the calculations is kept constant for all specimens, while the structural mass increases with infill density, the ratio $E/M$ decreases, leading to lower calculated frequencies. Therefore, the trend of the calculated frequencies is negatively correlated with the variation in infill density.

On the other hand, the experimental modal results reveal that the measured natural frequencies of the three materials change only slightly with different infill densities. This is an interesting phenomenon, because intuitively, higher infill ratios

**Table 2. Calculated and measured frequencies.**

| No. | PLA | | | PETG | | | ASA | | |
|---|---|---|---|---|---|---|---|---|---|
| | 5% | 15% | 30% | 5% | 15% | 30% | 5% | 15% | 30% |
| 1st Order Calc. Freq. | 42.18 | 40.31 | 37.13 | 41.94 | 39.84 | 36.75 | 43.39 | 41.07 | 38.47 |
| 1st Order Meas. Freq. | 44.02 | 43.93 | 43.91 | 39.07 | 39.04 | 39.02 | 43.97 | 43.92 | 43.89 |
| Error (%) | −4.18 | −8.24 | −15.44 | +7.35 | +2.05 | −5.82 | −1.32 | −6.49 | −12.35 |
| 2nd Order Calc. Freq. | 264.75 | 252.09 | 231.25 | 263.06 | 248.94 | 228.85 | 273.06 | 257.23 | 239.94 |
| 2nd Order Meas. Freq. | 270.99 | 266.11 | 258.79 | 244.61 | 234.38 | 214.84 | 261.23 | 258.79 | 239.26 |
| Error(%) | −2.30 | −5.27 | −10.64 | +7.54 | +6.21 | +6.52 | +4.53 | −0.60 | +0.28 |

Explanation:

1st Order Calc. Freq.: Calculated first-order frequency.

1st Order Meas. Freq.: Measured first-order frequency.

Error (%): Percentage error between calculated and measured frequencies.

2nd Order Calc. Freq.: Calculated second-order frequency.

2nd Order Meas. Freq.: Measured second-order frequency.

should increase the elastic modulus or stiffness of the structure. However, the experimental results suggest otherwise. From the same relation, $f \propto \sqrt{E/M}$, it can be seen that as the infill ratio increases, both the elastic modulus and the structural mass increase. When the rates of increase are comparable, the ratio $E/M$ remains nearly constant, and thus the natural frequency does not vary significantly. This conjecture will be further examined and confirmed in subsequent sections based on the identified elastic moduli.

Comparing the initially calculated frequencies with the experimentally measured values for the three materials (PLA, PETG, and ASA) reveals the frequency errors corresponding to the three infill densities (5%, 15%, and 30%). The maximum error in the first-order frequency is −15.44%, while that in the second-order frequency is −10.6%. These results indicate that the elastic modulus assumed in the initial FE model is lower than that of the actual specimens, thus requiring model updating.

From a materials perspective, it can be observed that at a 5% infill rate, components printed with PLA exhibit the highest measured frequencies, while those printed with PETG show the lowest, with a notable difference of 12.67%. At a 15% infill rate, the measured frequency of PLA components remains the highest, whereas the frequencies of the other two materials are nearly equivalent. At a 30% infill rate, the measured frequencies of all three materials become almost indistinguishable. This result indicates that the nominal elastic modulus of components printed with different materials is significantly influenced by the infill rate, leading to varying dynamic responses.

Fig 6 illustrates the measured mode shapes of the samples compared to the calculated mode shapes from the initial model. Significant differences in modal shapes can be clearly observed among samples with different infill rates and material types. Even for components with closely related measured frequencies, their measured mode shapes display considerable variations. This indicates that mode shapes provide a more accurate reflection of internal structural differences, revealing underlying mechanisms such as localized stiffness distribution, material heterogeneity, and the impact of infill parameters. Therefore, relying solely on measured frequencies as an identification metric makes it challenging to accurately discern the true parameters of 3D-printed components, potentially leading to misjudgments regarding their mechanical behavior. This underscores the indispensable role of integrating frequency and mode shape information in accurately elucidating the relationships between 3D printed material parameters and structural dynamics, thereby enhancing identification accuracy.

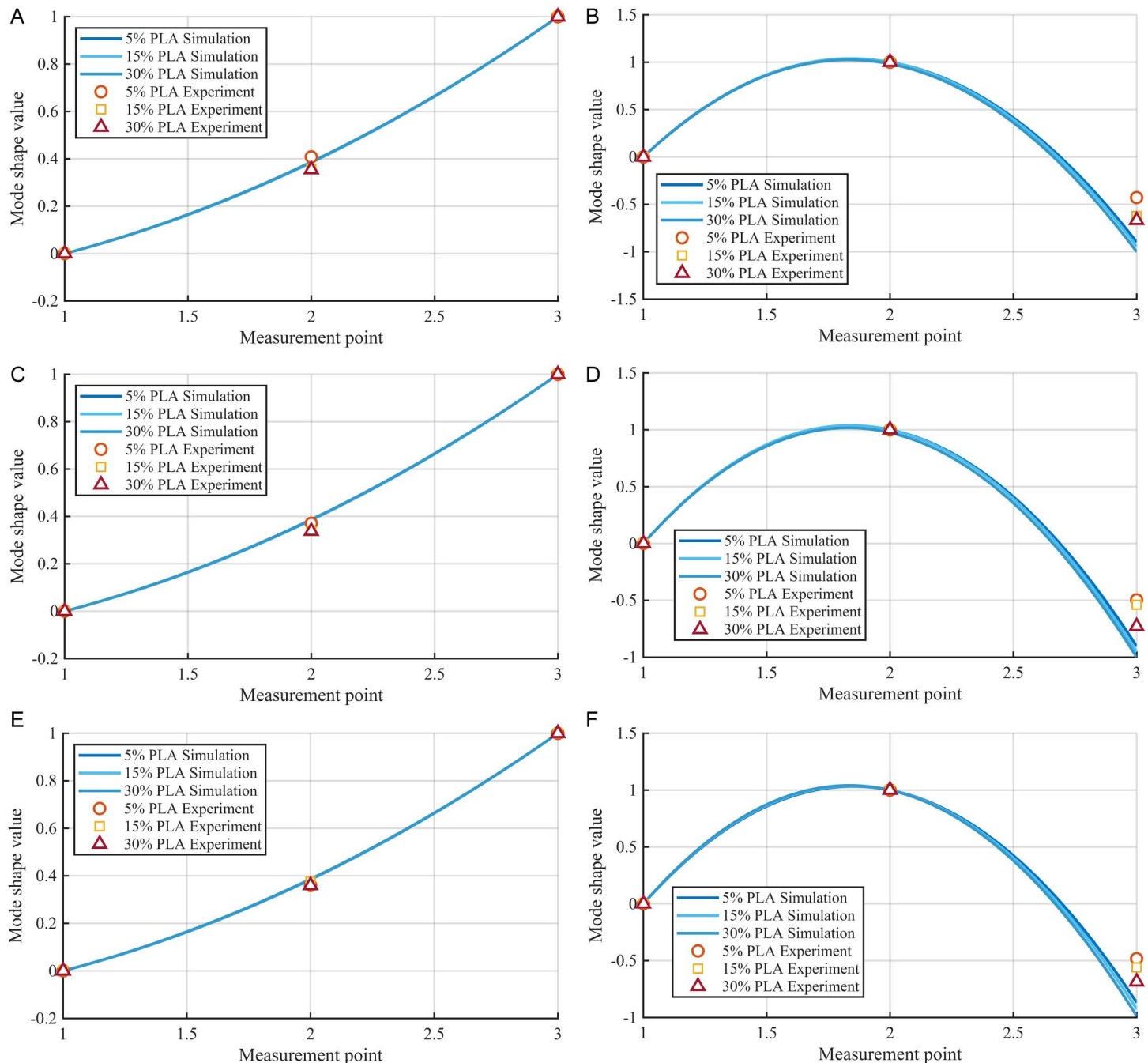

**Fig 6. Modal shape of each sample.** (a) 1st Mode Shape of PLA, (b) 2nd Mode Shape of PLA, (c) 1st Mode Shape of PETG, (d) 2nd Mode Shape of PETG, (e) 1st Mode Shape of ASA, (f) 2nd Mode Shape of ASA.

## 3.4 Finite element model updating

Considering the notable differences between the measured frequencies and mode shapes and their previously calculated values, the updating of the nominal elastic modulus for the nine 3D-printed components will follow the procedure outlined in Fig 1. The data used for model updating includes the first two measured frequencies and the first two mode shapes

from the two measurement points shown in Fig 7. The updating factor is defined as $\theta = E_r/E_0$, where $E_r$ denotes the actual elastic modulus of the structure and $E_0$ is the nominal value. Based on the selected response indices, a RF surrogate model is first constructed. Taking the PLA specimen with a 30% infill ratio as an illustrative example, a total of N = 600 elastic modulus samples were generated within [400,900] MPa using Latin hypercube sampling [43]. For each sample, the FE model provided the first two natural frequencies and two nodal values of the mode shapes, which were used as outputs for training four separate RF regressors. The dataset was split into 500 training and 100 test samples with a fixed random seed. To improve robustness to measurement perturbations while preserving unbiased targets, a 1% multiplicative Gaussian noise was applied to the output frequencies of 50% of the training samples (semi-noise augmentation), while inputs and test data remained noise-free. The RF models were implemented with MATLAB's standard fitrensemble function (Method = 'Bag'), using B = 300 regression trees with bootstrap aggregation of CART base learners. No explicit depth limit was enforced, and the minimum leaf size was kept at the default value; these settings were chosen after a sensitivity analysis indicated negligible gains beyond B ≈ 300B. Model accuracy was evaluated on the independent test set using the Pearson correlation coefficient (R) [38] and mean squared error (MSE) [39].

Fig 8 illustrates the prediction accuracy of the first two natural frequencies for the PLA specimen with a 30% infill ratio. As shown in Fig 8, the frequencies predicted by the RF model are in excellent agreement with those obtained from the FE analysis, confirming the effectiveness of the RF surrogate in approximating the modal responses of this specimen.

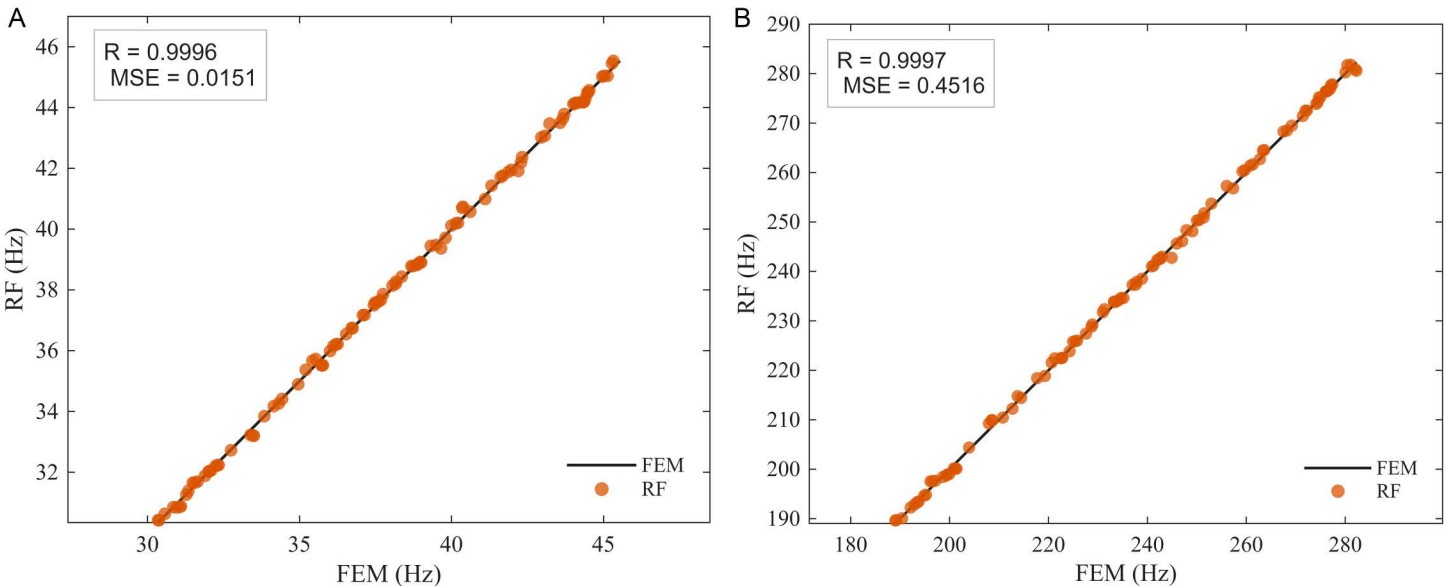

**Fig 7. Schematic of boundary conditions and sensor locations used for the beam.**

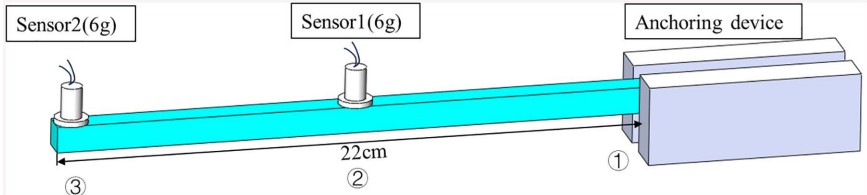

**Fig 8. Prediction of natural frequencies by the RF surrogate model for the PLA specimen with 30% infill ratio. (a)** First natural frequency, **(b)** Second natural frequency.

Subsequently, Eqs. (1)–(5) were employed to construct the Markov chains for the updating factors of each component and to estimate the mean values of their elastic moduli. The number of samples for the Markov chain is set to 10,000. As a result, the elastic moduli of the nine components are obtained, as shown in Table 3. For clarity, the values in Table 3 are posterior means identified in this study from our experimental modal measurements by the Bayesian updating procedure described in Section 2.1; they are not taken from prior publications. The nominal modulus $E_0 = 600\,MPa$ was selected as a baseline to non-dimensionalize the correction factor $\theta$ and to initialize the FE model; it is an assumed starting value and is not reported as a result.

From Table 3, it is evident that the updated elastic moduli of samples with different materials and infill rates exhibit significant differences. As the infill rate increases, the elastic moduli of the three materials show a consistent upward trend. For example, in the case of PLA, the identified elastic moduli at infill rates of 5%, 15%, and 30% are 634 MPa, 716 MPa, and 793 MPa, respectively, demonstrating a clear increasing relationship. The elastic moduli of PETG and ASA materials show a similar trend, with the values for PETG being 475 MPa, 553 MPa, and 608 MPa at 5%, 15%, and 30% infill rates, respectively, and those for ASA being 580 MPa, 644 MPa, and 607 MPa for the same infill densities. These trends confirm the strong dependency of the elastic modulus on the infill density for all three materials [40].

To clearly present the updated elastic modulus results, Fig 9 provides the bar chart of the elastic moduli for the nine components. As shown in Fig 9, when compared at the same infill ratio, PLA consistently exhibits the highest modulus and PETG the lowest. At 5% infill, the modulus of PLA is 29.2% higher than that of PETG; at 15% infill, the difference remains 23.2%; and at 30% infill, PLA exceeds PETG by 32.3%. These relative gaps are consistent with the well-established monotonic increase of bending stiffness with relative density in FDM parts and with commonly reported material rankings under comparable printing conditions, whereby PLA typically exhibits a higher glassy modulus and stronger interroad bonding than PETG and ASA [41]. Within the 5–30% infill range considered here, the effective modulus is governed by both the intrinsic polymer stiffness and the cellular-solid scaling of the printed architecture; increasing infill reduces void fraction and shear-lag and therefore yields larger stiffness increments, particularly for PLA. From a design standpoint, the observed 20–30% modulus advantage of PLA over PETG at fixed infill suggests that material selection can deliver stiffness gains comparable to those obtained by increasing infill from 15% to 30%, with only a relatively small increase in mass.

These observed relative differences are comparable to the 20%−25% growth in bending elastic modulus reported in literature [41] through three-point bending tests under static loading when infill density increased from 50% to 75%. Furthermore, by comparing the relative increases in elastic modulus and mass (with respect to the 5% infill case), we identified the underlying reason why the measured frequencies of all specimens change only slightly with infill. Taking PLA as an example, when the infill ratio increases from 5% to 30%, the total mass—including the 12 g sensor—shows a growth rate of 40.94%, whereas the elastic modulus increases by only 23.8%. According to the dynamic relation $f \propto \sqrt{E/M}$, this corresponds to an estimated frequency reduction of approximately −8.6%, which well explains why the measured natural

**Table 3. Updated results of parameters (posterior means identified from experimental modal data).**

| No. | PLA | | | PETG | | | ASA | | |
|---|---|---|---|---|---|---|---|---|---|
| | 5% | 15% | 30% | 5% | 15% | 30% | 5% | 15% | 30% |
| θ | 0.80 | 0.86 | 0.99 | 0.65 | 0.69 | 0.74 | 0.72 | 0.80 | 0.84 |
| $E_r$ (MPa) | 634 | 716 | 793 | 475 | 553 | 608 | 580 | 644 | 607 |

Explanation:

θ denotes the dimensionless updating factor used in the stiffness-updating formulation.

$E_r$ (MPa) denotes the posterior mean elastic modulus identified from the measured modal data (first two modes) using the Bayesian updating described in Section 2.1.

The nominal modulus $E_0 = 600\,MPa$ was chosen as an initial baseline to non-dimensionalize $\theta$ and initialize the FE model; it is not a literature value.

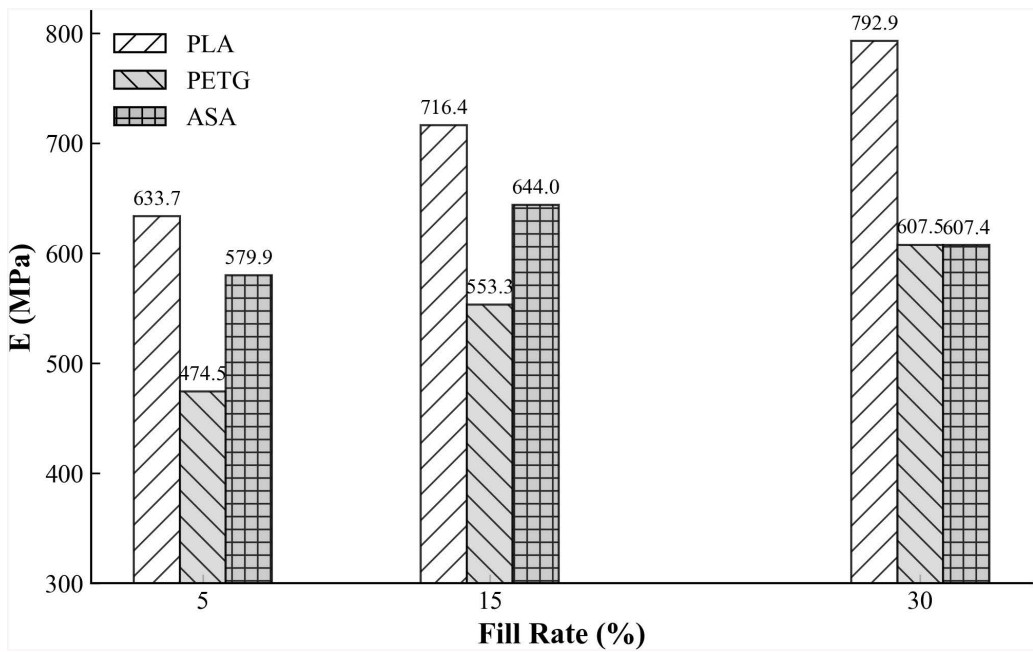

**Fig 9. Updated elastic modulus of each material.**

frequencies decrease slightly with increasing infill ratio. It should be noted, however, that a more refined prediction would require accounting for the fact that the sensor mass is positioned at the mid-span and free end of the cantilever, which influences the distribution of mass and thus the measured frequencies.

In terms of material types, the growth rate of the elastic modulus for PLA components increases by 32.5% as the infill rate rises from 5% to 30%. For PETG components, the growth rate is 9.2%, and for ASA components, it is 19.2%. These results indicate that PLA demonstrates the best mechanical performance among the materials tested.

Furthermore, the updated elastic moduli were used to calculate the frequencies of all components and determine the relative deviations from the measured results, as shown in Fig 10. From Fig 10, it is evident that after model updating, the relative errors between the calculated frequencies and experimental frequencies for the three materials at different infill rates and modes are significantly reduced. Before updating, the deviation between the simulated frequencies and experimental frequencies ranged from 10% to 30%. After updating, most components exhibit relative errors within 3%, with some samples even below 1%. For example, for PLA at a 30% infill rate, the first-order frequency error decreased from −15.44% to −2.837%, while the second-order frequency error reduced from −10.64% to 2.685%. The errors for groups of PETG and ASA materials are mostly controlled to within 3%. The error for ASA at a 30% infill rate is slightly larger, with a first-order frequency error of −7.15% and a second-order frequency error of 6.248%, which is very close to the magnitude of deviation in the elastic modulus identified in literature [42]. This unavoidable deviation can be reasonably attributed to measurement noise and slight differences in boundary conditions at the clamp. The remaining discrepancy of about 2–3% after updating is consistent with uncertainty propagation. Because the natural frequency scales approximately with the square root of the stiffness-to-mass ratio, a 4–6% spread in elastic modulus for comparable printed coupons—commonly reported in the literature [42]—leads to an expected frequency variation of roughly 2–3% when mass is held constant. Minor deviations from an ideal clamped boundary and the added mass and local compliance of the 12 g sensor can introduce additional, mode-dependent shifts of about 1%, with the first mode being most sensitive near the clamp. Together with

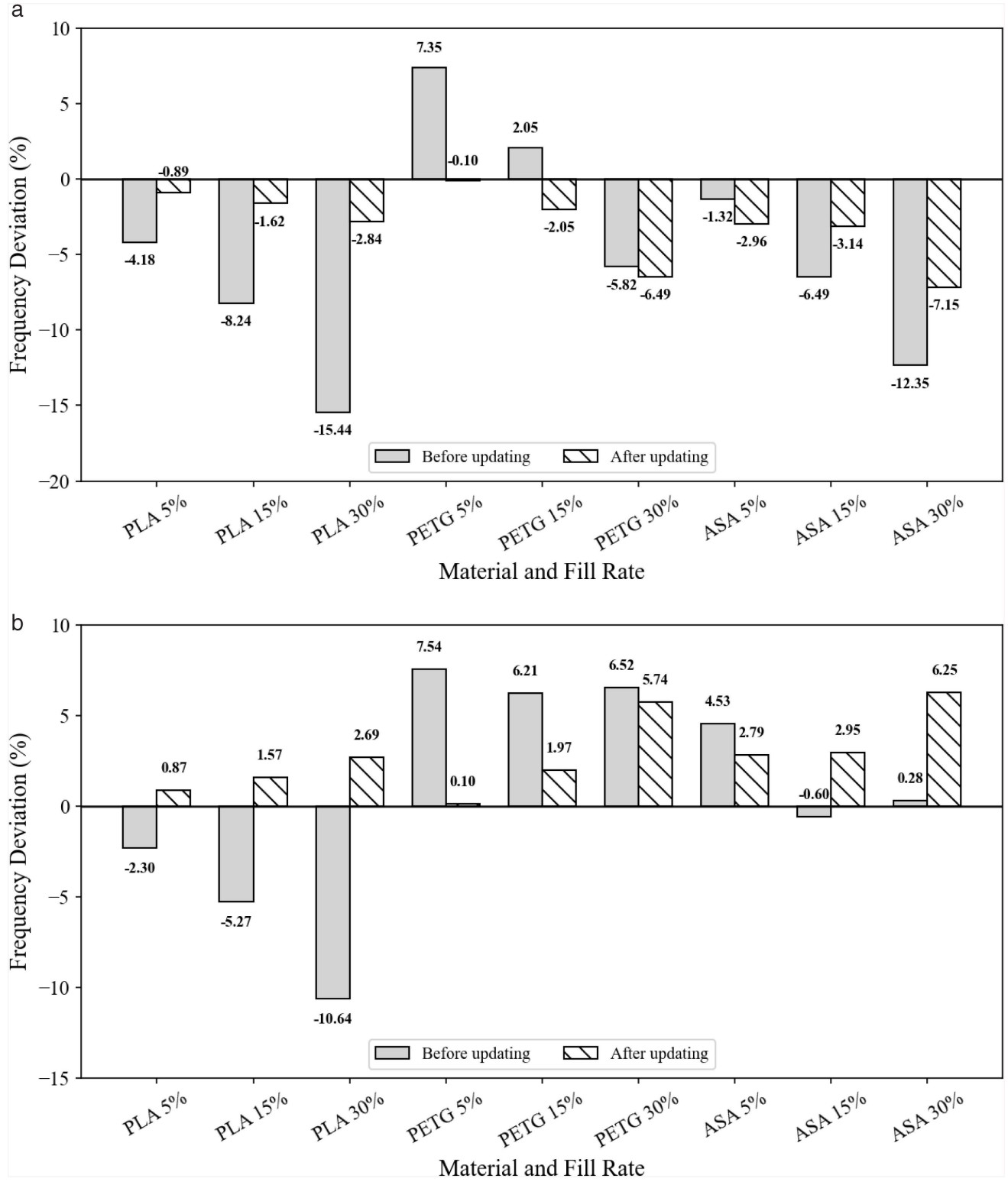

**Fig 10. Frequency errors before and after updating.** (a) 1st-order Frequency, (b) 2nd-order frequency.

measurement noise and specimen-to-specimen variability, these factors account for the small material-dependent asymmetries that persist in Fig 10. The consistently high MAC values in Table 4 (exceeding 0.98 and typically around 0.99) indicate that the updated model reproduces the measured mode shapes well, suggesting that the residual differences are mainly due to slight stiffness-amplitude bias rather than discrepancies in the mode-shape pattern. The additional mass of the sensor and small variations in clamping stiffness are sufficient to explain the observed frequency shift. These results clearly demonstrate that the updated elastic moduli of the components accurately reproduce their frequencies, validating the effectiveness of the updating process.

Finally, let's examine the updated mode shapes. Table 4 presents the first-order and second-order MAC (Modal Assurance Criterion) values for all material and infill rate groups. From Table 4, it can be seen that the MAC values for all nine components are greater than 0.98, with the majority reaching 0.99 or even 0.999. This result indicates that the updated FE model can accurately reproduce the true mode shapes of the structures, ensuring that the dynamic response of the model remains consistent with the actual structure. It further demonstrates that the updating results are also highly accurate in predicting mode shapes.

## 3.5 Discussion

This study successfully introduced a non-destructive framework for identifying the elastic modulus of FDM 3D-printed components. The method effectively integrates FE models, RFSM, and Bayesian inference to update material parameters, achieving high accuracy in predicting modal frequencies and mode shapes.

The results show that the elastic modulus of 3D-printed components is highly sensitive to infill density, with PLA exhibiting the most significant increase in modulus as infill density rises, followed by ASA and PETG. The RFSM provided accurate predictions, with frequency errors largely within 3%, and MAC values consistently exceeding 0.99, confirming the model's effectiveness.

However, the current study assumes that the material is isotropic in the vertical bending direction, which is a reasonable approximation given the uniform infill in this direction. Future work could consider incorporating an anisotropic material model to capture direction-dependent properties, particularly for more complex geometries. Moreover, more precise sensor calibration and measurement noise mitigation could improve the accuracy of the modal testing results.

The method proposed here is particularly valuable for industries requiring reliable, non-destructive methods to assess the mechanical properties of 3D-printed components, such as aerospace and automotive sectors. Future improvements could include extending the framework to handle more complex geometries and multi-scale material models.

## 4. Conclusions

This paper presented and verified an efficient, non-destructive Bayesian framework for parameter updating, which integrates FE models, a RF surrogate model, and probabilistic inference to calibrate the equivalent elastic modulus of FDM 3D-printed components.

**Table 4. Updated MAC values.**

| MAC value | PLA | | | PETG | | | ASA | | |
|---|---|---|---|---|---|---|---|---|---|
| | 5% | 15% | 30% | 5% | 15% | 30% | 5% | 15% | 30% |
| 1st order | 0.9995 | 0.999 | 0.9989 | 0.9994 | 0.9984 | 0.9983 | 0.9991 | 0.9993 | 0.999 |
| 2nd order | 0.993 | 0.992 | 0.9938 | 0.9911 | 0.9921 | 0.9941 | 0.9911 | 0.9925 | 0.9947 |

Main results and findings:

- **Specimen design and testing:** Nine cantilever beam specimens were fabricated using three materials (PLA, PETG, ASA) and three infill densities (5%, 15%, 30%). Operational modal analysis successfully obtained the first two natural frequencies and mode shapes.

- **Surrogate modeling:** A RF surrogate model was trained to approximate the nonlinear relationship between elastic modulus and modal responses, significantly reducing the computational cost of Bayesian MCMC-based updating.

- **Parameter updating:** Bayesian inference combined with RF surrogate model yielded the posterior distributions of equivalent elastic moduli for all specimens.

- Key quantitative results:

  (1)  PLA exhibited the strongest dependence on infill density, with a 29.2% increase in modulus between 5% and 30% infill.

  (2)  Frequency prediction errors of the updated FE model were largely within 3%, and all modal assurance criterion (MAC) values exceeded 0.99, confirming high accuracy and physical consistency.

Contributions and implications:

- Demonstrated that the equivalent elastic modulus of FDM components is a macro-scale property dependent on infill density, rather than a fixed intrinsic material constant.

- Provided a non-destructive method for directly calibrating the mechanical performance of printed components at the structural level, eliminating the need for destructive coupon tests.

- The proposed framework offers reliable data for quality control, performance prediction, and design optimization, and is extendable to more complex geometries, other additive manufacturing processes (e.g., SLA, SLS), and additional physical parameters such as damping and anisotropy.

- Future work will explore integration into automated quality inspection and multi-scale modeling to establish digital closed-loops from design and manufacturing to performance verification.

## Supporting information

**S1 File. Processed data for figures.**
(XLSX)

**S2 File. Raw experimental data.**
(XLSX)

## Author contributions

**Conceptualization:** Jin Zhang, lili Lu, Ting Zhu.

**Data curation:** Jin Zhang, lili Lu, Ting Zhu.

**Formal analysis:** Jin Zhang, lili Lu, Ping Feng.

**Investigation:** Jin Zhang, lili Lu, Ping Feng, Ting Zhu.

**Methodology:** Jin Zhang, lili Lu.

**Project administration:** Jin Zhang, Ping Feng.

**Resources:** Jin Zhang, lili Lu.

**Software:** Jin Zhang, Ting Zhu.

**Supervision:** Jin Zhang, lili Lu, Ping Feng, Ting Zhu.

**Validation:** lili Lu.

**Visualization:** lili Lu.

**Writing – original draft:** Jin Zhang.

**Writing – review & editing:** Jin Zhang, lili Lu, Ping Feng, Ting Zhu.

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
