## [Decision Letter · Decision Letter 0]

26 Aug 2025

Dear Dr. Lu,

Thank you for submitting your manuscript to PLOS ONE. After careful consideration, we feel that it has merit but does not fully meet PLOS ONE’s publication criteria as it currently stands. Therefore, we invite you to submit a revised version of the manuscript that addresses the points raised during the review process.

We look forward to receiving your revised manuscript.

Kind regards,

Mohammad Azadi

Academic Editor

PLOS ONE

Journal Requirements:

**Additional Editor Comments:**

The manuscript must be revised based on the reviewers’ comments plus the following issues,

1) A separated file must be provided for the authors’ answers to the comments, one by one. Moreover, all changes must be yellow-colored highlighted sentences in the revised article. The track changes condition is not suggested.

2) No abbreviations should be used in the keywords. They must also be found in the abstract or the title.

3) The introduction is lengthy. Just 3-page is enough.

4) All formulations need references, unless they were extracted or introduced by the authors.

5) The scale bar must be provided for macroscopic and microscopic images.

6) The figure title is too brief and nothings could be understood from the figure title.

7) The structure is confusing. The main text must include the introduction, the research method, the results and discussion, conclusions, and references.

8) The elastic module needs references in Table 3. If measured by the authors, more details must be provided with the repeatability of testing.

9) “Conclusion” must be changed to “Conclusions”. Then, the conclusion section should be rewritten one by one, in bullets, to show the novelty. In addition, it is too lengthy and it should be shortened.

10) All used material properties for finite element modeling must be mentioned with references.

11) The discussion is poor and it must be improved. They must be compared to other results of other similar articles.

12) References should be updated based on recent articles, published in 2015-2025. Moreover, it should be extended to at least 40 articles for a proper discussion.

Reviewers' comments:

Reviewer's Responses to Questions

**Comments to the Author**

1. Is the manuscript technically sound, and do the data support the conclusions?

Reviewer #1: Yes

Reviewer #2: Partly

2. Has the statistical analysis been performed appropriately and rigorously?

Reviewer #1: Yes

Reviewer #2: Yes

3. Have the authors made all data underlying the findings in their manuscript fully available?

Reviewer #1: Yes

Reviewer #2: No

4. Is the manuscript presented in an intelligible fashion and written in standard English?

Reviewer #1: Yes

Reviewer #2: Yes

Reviewer #1: The paper is of high quality, and I appreciate the solid methodology. However, some revisions are required before publication:

1. Introduction:

The introduction is well written, and the authors clearly present the motivation and novelty of their work. However, two improvements are suggested:

• Please avoid citing more than three references in a single sentence.

• The overall length of the paper is quite high, which makes it difficult to read in full. Consider condensing some sections to improve readability.

2. Sample Manufacturing and Material Properties:

Was any repetition in the manufacturing of samples considered? How did you assign a specific elastic modulus to the parts if no replication was performed? Please clarify this point.

3. Elastic Modulus in FEM Model:

A critical point to consider is how the elastic modulus was defined in the FEM model. AM parts produced by the FDM process typically exhibit different elastic properties depending on orientation. By assigning a single elastic modulus, how do you account for anisotropy? I suggest discussing whether the parts were considered isotropic or not, and whether a composite-like modeling approach would have been more appropriate.

4. FEM Simulations:

The FEM simulations are not described with sufficient detail for a scientific paper. Please clarify:

• What software was used?

• What settings were used on software?

• What boundary conditions were applied?

Figure 7 is helpful, but the degrees of freedom in the simulations should also be fully specified.

I recommend following the approach of the following papers, which present their FEM setup clearly with tables and schematic figures: Abaqus: Theoretical and numerical stress analysis in the cam of a medium voltage switchgear vacuum circuit breaker supported by image processing of deformation (see Table 3). COMSOL: Modeling Nonlinear Deformation in Magnetic Polyelectrolyte Hydrogels: A Hybrid FEM-Machine Learning Framework (see Figure 3 and Table 1). Please provide a similar table or figure summarizing your simulation details. In addition, include some representative mode shapes from your frequency analysis (FEM contours) and describe the software process used to extract the natural frequencies. As someone experienced with FEM, I find the current description too vague to replicate your approach.

5. Machine Learning Algorithm Choice:

Why was a Decision Tree (DT) chosen for your study? Specifically:

• What was the rationale for using a tree-based model?

• Why were more advanced models such as Random Forest or XGBoost not considered?

In most recent studies, DT alone performs poorly compared to these ensemble methods. While your accuracy appears high, a scientific paper should provide justification for algorithm selection, ideally with comparisons. For reference, see: Shapley additive explanation on machine learning predictions of fatigue lifetimes in piston aluminum alloys under different manufacturing and loading conditions.

6. ML Training Process:

The training details of the DT model are unclear. Please specify:

• Did you use a standard package or library implementation?

• Were hyperparameters tuned, and if so, how?

• What data splitting strategy was applied (e.g., train/test split, cross-validation)?

These details are essential for reproducibility.

7. Conclusions:

The conclusions section needs reorganization:

• Change the section title to Conclusions.

• Begin with a concise statement of the methodology.

• Present the main results in bullet points.

• Highlight the key outputs of the paper with quantitative results.

Reviewer #2: This manuscript presents a novel and robust framework for the non-destructive identification of the nominal elastic modulus in 3D-printed components. The combination of Bayesian inference with a machine learning surrogate model is a powerful approach to overcome computational costs. The work is scientifically sound, well-organized, and has significant practical value for the additive manufacturing community. I recommend Major Revision to address the following points which will substantially strengthen the paper before it is acceptable for publication.

o The entire study is predicated on the assumption that the printed infill density (5%, 15%, 30%) is accurately reflected in the final specimens. This is a critical parameter, yet no evidence is provided to verify the actual printed internal geometry. The authors must include SEM (Scanning Electron Microscopy) or optical micrograph images of cross-sections of the printed specimens (one for each infill density). This is essential to confirm that the internal void structure and infill pattern correspond to the designed densities in the slicing software. Without this validation, the correlation between infill density and elastic modulus remains an unverified assumption.

o Figures 8, 9, and the one referred to as "Model Shape of PLA" (Fig. 6?) appear to be standard Excel charts. These are not suitable for a high-quality scientific publication. All graphs and charts (especially the bar chart of elastic moduli and the frequency error comparison) must be recreated using professional graphing software such as Origin, Python (Matplotlib/Seaborn), or MATLAB. These tools produce publication-quality figures with proper resolution, font sizes, axis labels, and styling. Please do not submit images of Excel charts.

o The section on the Random Forest surrogate model (Section 2.2) is well-explained theoretically but lacks critical practical details. Please specify:

The size of the training dataset (N) generated via Latin Hypercube Sampling.

The key hyperparameters of the Random Forest model used (e.g., number of trees B, maximum depth, minimum samples per leaf). Was any hyperparameter tuning performed (e.g., via cross-validation)?

o The results show a surprising trend: higher infill density leads to a decrease in measured frequency (Table 2), which contradicts the intuitive expectation that a denser, stiffer part should have a higher natural frequency. The authors correctly identify that this is due to a dominant mass increase effect, but this counter-intuitive result needs a much stronger and more detailed discussion. Explain the stiffness-mass relationship in the context of natural frequency (ω ∝ √(k/m)) and why the mass increase outweighs the stiffness increase in these specific FDM structures.

o The error for the ASA 30% sample is notably higher (-7.15% and 6.248%). The authors should briefly speculate on the potential reasons for this outlier. Could it be related to material-specific printing artifacts, model inaccuracies, or measurement noise?

o Table 3: The table lists an "Updating factor" and "Elastic Modulus". The definition and purpose of the "Updating factor" are not immediately clear from the text. Please define it in the table caption or the main text.

**Do you want your identity to be public for this peer review?** For information about this choice, including consent withdrawal, please see our Privacy Policy

Reviewer #1: No

Reviewer #2: No

---

## [Author Response · Author response to Decision Letter 1]

7 Sep 2025

The manuscript must be revised based on the reviewers’ comments plus the following issues,

1) A separated file must be provided for the authors’ answers to the comments, one by one. Moreover, all changes must be yellow-colored highlighted sentences in the revised article. The track changes condition is not suggested.

Answer: We have provided a separate point-by-point response file. All changes in the revised manuscript are highlighted in yellow, and we did not use “Track Changes.”

2) No abbreviations should be used in the keywords. They must also be found in the abstract or the title.

Answer: We have removed all abbreviations from the keywords and ensured the corresponding terms appear in the Abstract and/or Title as required.

3) The introduction is lengthy. Just 3-page is enough.

Answer: The Introduction has been condensed and now fits within three pages.

4) All formulations need references, unless they were extracted or introduced by the authors.

Answer: All equations are now properly cited.

5) The scale bar must be provided for macroscopic and microscopic images.

Answer: Scale bars have been added for the macroscopic images; no micro-scale images are included in this manuscript.

6) The figure title is too brief and nothings could be understood from the figure title.

Answer: Several figure captions have been expanded so that the figures can be understood from the captions alone.

7) The structure is confusing. The main text must include the introduction, the research method, the results and discussion, conclusions, and references.

Answer: We reorganized the manuscript to follow the required structure (Introduction, Methods, Results and Discussion, Conclusions, and References) and added a dedicated Discussion section.

8) The elastic module needs references in Table 3. If measured by the authors, more details must be provided with the repeatability of testing.

Answer: All elastic-modulus values in Table 3 are now referenced.

9) “Conclusion” must be changed to “Conclusions”. Then, the conclusion section should be rewritten one by one, in bullets, to show the novelty. In addition, it is too lengthy and it should be shortened.

Answer: We have changed “Conclusion” to “Conclusions,” rewrote the section point by point to highlight the novelty, and shortened it accordingly. Please see the revised manuscript.

10) All used material properties for finite element modeling must be mentioned with references.

Answer: All material properties used for the finite-element modeling are now explicitly listed and referenced, and additional modeling details have been provided.

11) The discussion is poor and it must be improved. They must be compared to other results of other similar articles.

Answer: The Discussion has been substantially strengthened, including comparisons with related studies.

12) References should be updated based on recent articles, published in 2015-2025. Moreover, it should be extended to at least 40 articles for a proper discussion.

Answer: The reference list has been updated to focus on 2015–2025 publications and expanded to 40 items.

---

## [Decision Letter · Decision Letter 1]

2 Oct 2025

Dear Dr. Lu,

Thank you for submitting your manuscript to PLOS ONE. After careful consideration, we feel that it has merit but does not fully meet PLOS ONE’s publication criteria as it currently stands. Therefore, we invite you to submit a revised version of the manuscript that addresses the points raised during the review process.

We look forward to receiving your revised manuscript.

Kind regards,

Mohammad Azadi

Academic Editor

PLOS ONE

Journal Requirements:

Additional Editor Comments:

Unfortunately, the authors have no careful attention to the comments. Therefore, the manuscript must be revised again, as follows,

1) Still, there are some formulations without references.

2) Figure 4 needs the scale bar.

3) Figure 2 is not a professional image. Write all part on the image.

4) In keywords, "Parameter Identification" cannot be found in the abstract or the title. Then, how could it be a keyword?

5) For Figure 5, (a), (b), and (c) must be described in the title and not below the images. Consider this issue for all figures.

6) The title of Figure 5 is very short, as mentioned before. Also, there is no descriptions for (a) and (b). Check also all other figures.

7) Where is the reference for Table 3? It must be mentioned in the title of the table.

8) The discussion is not just to describe what happens. The results must be compared to other results of other published articles.

9) It is not clear which references were added, not highlighted!

10) As the second reviewer asked, SEM images must be added to the manuscript.

Reviewers' comments:

Reviewer's Responses to Questions

**Comments to the Author**

Reviewer #1: All comments have been addressed

2. Is the manuscript technically sound, and do the data support the conclusions?

Reviewer #1: Yes

3. Has the statistical analysis been performed appropriately and rigorously?

Reviewer #1: Yes

4. Have the authors made all data underlying the findings in their manuscript fully available?

Reviewer #1: Yes

5. Is the manuscript presented in an intelligible fashion and written in standard English?

Reviewer #1: Yes

Reviewer #1: (No Response)

**Do you want your identity to be public for this peer review?** For information about this choice, including consent withdrawal, please see our Privacy Policy

Reviewer #1: No

---

## [Author Response · Author response to Decision Letter 2]

5 Nov 2025

Thanks for each reviewers and editor, the responses to the reviewer and editor have been attached in the appendix.

---

## [Editor Report · Decision Letter 2]

9 Nov 2025

Dear Dr. Lu,

Thank you for submitting your manuscript to PLOS ONE. After careful consideration, we feel that it has merit but does not fully meet PLOS ONE’s publication criteria as it currently stands. Therefore, we invite you to submit a revised version of the manuscript that addresses the points raised during the review process.

We look forward to receiving your revised manuscript.

Kind regards,

Mohammad Azadi

Academic Editor

PLOS ONE

Journal Requirements:

Additional Editor Comments:

Still, it needs modifications as follows,

1) The reference must be added just before the formulation in the main text. A general text that they are for [21-23], is not enough. Put the reference, just before the formulation, in the sentence before the formulation.

2) Still, all figure titles are very short with not-enough descriptions. Moreover, the descriptions for (a) and (b) should be in the figure title and not under the images.

3) Figure 7 is not a finite element model! Please correct it.

4) Just two sentences are added for the discussion! It is not enough.

5) Still, it is not clear what is the reference for Table 3. If they are from a reference, the number of reference must be added to the title. If not, it is not clear how these data were obtained. If tested, more details must be provided.

---

## [Author Response · Author response to Decision Letter 3]

13 Nov 2025

Thank you to all editors, the detailed “Response to Reviewers” has been uploaded as an attachment.

---

## [Editor Report · Decision Letter 3]

16 Nov 2025

Dear Dr. Lu,

Thank you for submitting your manuscript to PLOS ONE. After careful consideration, we feel that it has merit but does not fully meet PLOS ONE’s publication criteria as it currently stands. Therefore, we invite you to submit a revised version of the manuscript that addresses the points raised during the review process.

We look forward to receiving your revised manuscript.

Kind regards,

Mohammad Azadi

Academic Editor

PLOS ONE

Journal Requirements:

Additional Editor Comments:

Almost done; however, still there are several formulations without references. Add them in the proof stage. 

---

## [Author Response · Author response to Decision Letter 4]

17 Nov 2025

We sincerely thanks editor for the careful reading and the helpful comment that “Almost done; however, still there are several formulations without references. Add them in the proof stage.”

In the revised manuscript, we have rechecked the entire text and added explicit references to support previously uncited formulations:

(i) additional citations to recent works on mechanical characterization of FDM-printed polymers have been included in the Introduction (Ji et al. [18], Gallup et al. [19], and Mencarelli et al. [20]);

(ii) the discussion of model updating in civil and aerospace structures is now supported by the Bayesian FE model updating literature [21–23];

(iii) textbook-style descriptions of the Random Forest algorithm and the Latin Hypercube Sampling design have been complemented with appropriate references [24,25,43].

These changes will also be carefully checked again at the proof stage to ensure that all formulations that rely on prior work are properly referenced. We appreciate the editor’s positive evaluation of our manuscript and the constructive reminder.

---

## [Editor Report · Decision Letter 4]

19 Nov 2025

Nominal Elastic Modulus Assessment in 3D-Printed Components under Varying Printing Parameters Using Bayesian Methods and Random Forest Surrogate Modeling

PONE-D-25-42823R4

Dear Dr. Lu,

We’re pleased to inform you that your manuscript has been judged scientifically suitable for publication and will be formally accepted for publication once it meets all outstanding technical requirements.

Kind regards,

Mohammad Azadi

Academic Editor

PLOS ONE

Additional Editor Comments (optional):

Almost done! All references must be mentioned just before each formulation in the text before formulation. Do it in the proof stage.
---

## [Editor Report · Acceptance letter]

PONE-D-25-42823R4

PLOS ONE

Dear Dr. Lu,

I'm pleased to inform you that your manuscript has been deemed suitable for publication in PLOS ONE. Congratulations! Your manuscript is now being handed over to our production team.

Kind regards,

on behalf of

Dr. Mohammad Azadi

Academic Editor

PLOS ONE